



# Far ultraviolet airglow remote sensing measurements on Feng Yun 3D meteorological satellite

Yungang Wang[1,2],Liping Fu[3], Fang Jiang[3],Xiuqing Hu[1],Chengbao Liu[1], Xiaoxin Zhang[1,2],
Jiawei Li[1,2], Zhipeng Ren[4], Fei He[4], Lingfeng Sun[4], Ling Sun[1], Zhongdong Yang[1], Peng Zhang[1],
Jingsong Wang[1,2], Tian Mao[1,2]

1 National Satellite Meteorological Center, Chinese Meteorological Administration, Beijing, China

2 Key Laboratory of Space Weather, National Center for Space Weather, Chinese Meteorological Administration, Beijing, China

3 National Space Science Center,Chinese Academy of Sciences, Beijing, China

4 Key Laboratory of Earth and Planetary Physics, Institute of Geology and Geophysics, Chinese Academy of Sciences, Beijing, China

*Correspondence to*: *Tian Mao (email: maotian@cma.cn)*

**Abstract.** The Ionospheric Photometer (IPM) is carried on the Feng Yun 3D (FY3D) meteorological satellite, which allows for the measurement of far-ultraviolet (FUV) airglow radiation in the thermosphere. IPM is a compact and high-sensitivity nadir-viewing FUV remote sensing instrument. It monitors 135.6 nm emission in the night-side thermosphere and 135.6 nm and $N_2$ LBH emissions in the day-side thermosphere that can be used to invert the peak electron density of the F2 layer ($NmF_2$) at night and $O/N_2$ ratio in the daytime, respectively. Preliminary observations show that the IPM could monitor the global structure of the equatorial ionization anomaly (EIA) structure around 2:00 local time using OI 135.6 nm nightglow properly. It could also identify the reduction of $O/N_2$ in the high-latitude region during the geomagnetic storm of Aug. 26, 2018. The IPM derived $NmF_2$ accords well with that observed by 4 ionosonde stations along 120 °E with a standard deviation of 26.67%. Initial results demonstrate that the performance of IPM meets the designed requirement and therefore can be used to study the thermosphere and ionosphere in the future.

## 1 Introduction

The Earth's far-ultraviolet (FUV) airglow radiation from the thermosphere includes the emission of H, O, and $N_2$ and the absorption of $O_2$ (Meier, 1991). The OI 135.6 nm nightglow emission, which is mainly produced by the recombination of ionospheric $O^+$ and electron, can present spatial and temporal variations of the ionosphere in the nighttime. The 135.6 nm and $N_2$LBH dayglow emission, which are produced by energetic photon-electron impact excitation of the neutral atmosphere,





are used to derive the characteristics of column $O/N_2$ in the sunlit disk. The FUV radiation can be completely absorbed by the lower atmosphere, and the Earth's atmosphere is opaque to the FUV radiation. The background emission of FUV airglow

from the Earth's surface is absent. So FUV airglow radiation is particularly well-suited to space-based remote sensing (Paxton et al., 2003; Budzien et al., 2019). In past decades, FUV spectrography based on satellites has been used extensively in studying the thermosphere and ionosphere, such as, GUVI (the Global Ultra-Violet Imager) on the NASA TIMED (Thermosphere, Ionosphere, Mesosphere Energetics and Dynamics) satellite (Christensen et al., 2003), and the Far Ultraviolet Imager (FUV) on the NASA IMAGE (Imager for Magnetopause-to-Aurora Global Exploration) satellite (Sagawa

et al., 2005). The other useful equipment used in studying the thermosphere and ionosphere is ionospheric photometer, which is compact and high-sensitive. The U.S. Naval Research Laboratory firstly gave the concept for a new class of ionospheric photometer twenty years ago. It was supplied in the Tiny Ionospheric Photometer (TIP) on the Constellation Observing System for Meteorology, Ionosphere, and Climate satellites (Anthes et al., 2008; Dymond et al., 2016), complemented and upgraded in the Tiny Ionospheric Photometer (TIP) as part of the GPS Radio Occultation and Ultralviolet

Photometry –Colocated (GROUP-C) experience on the International Space Station (Budzien et al., 2019; Budzien et al., 2017), and notably improved in the Triple Tiny Ionospheric Photometer (Tri-TIP) in Coordinated Ionospheric Reconstruction  CubeSat Experience (Dymond et al., 2017; Stephan et al., 2018).

The compact and high-sensitivity nadir-viewing FUV Ionospheric Photometer (IMP) is one of ten scientific payloads aboard the Feng Yun 3D meteorological satellite. IPM monitors 135.6 nm emission in the night-side thermosphere and 135.6 nm

and $N_2$ LBH emissions in the day-side thermosphere by employing a filter wheel that adds two red-leak signal channels for daytime and nighttime red-leak respectively. Red-leak refers to weak residual sensitivity of the sensor to detect unwanted wavelengths including visible light that is "redder" than ultraviolet (Budzien et al., 2019). The main scientific objectives of IPM are follows: (1) Measure 135.6 nm emission in the night-side thermosphere to capture the large-scale structure of the low- and mid-latitude ionosphere. (2) Measure 135.6 nm and $N_2$ LBH emissions in the day-side thermosphere to capture

global variations $O/N_2$ ratio and evolutions of the thermosphere and ionosphere during extreme space weather events. The FY3D is an afternoon sun-synchronous satellite with an orbit altitude of 830 km, an inclination of 98.75 °and orbit period of ~102 minutes, and designed for weather forecast, atmospheric chemistry, climate change monitoring, and space weather monitoring. The FY3D satellite was launched at 18:35 UTC on November 14, 2017 from the Taiyuan Satellite Base, Shanxi province, China. This paper presents instrumental descriptions and initial observations by IPM.

## 55 2 Instrument Description

### 2.1 Instrument parameters requirements

According to the two main scientific objectives mentioned above, the IPM instrument parameters requirements are summarized in the Table1. In the design of the ionospheric photometer, there are two important problems to be solved. One problem is red-leak. It is a major challenge to ionospheric photometer that visible light radiation from the sun is about 109



times more than FUV radiation.  The other problem is that ionospheric photometers need to eliminate 130.44nm and shorter

wavelengths airglow and collect 135.6 nm airglow emission with high sensitivity.

**Table 1. FY-3D IPM instrument parameters requirements.**

| Parameter | value |
|---|---|
| Wavelength | 135.6 nm(night mode)<br>135.6 nm and 145-180nm(day mode) |
| Field of View | ~3.5°(along orbit)×1.6°(cross orbit) |
| Sensitivity | day mode:≥1 counts/s/Rayleigh@135.6nm<br>night mode:≥150 counts/s/Rayleigh@135.6nm |
| Spatial resolution | ~30km@ionosphere(300km) |
| Time resolution | 2 s(day mode)<br>10 s(night mode) |

## 2.2 Composition, channel, and mode

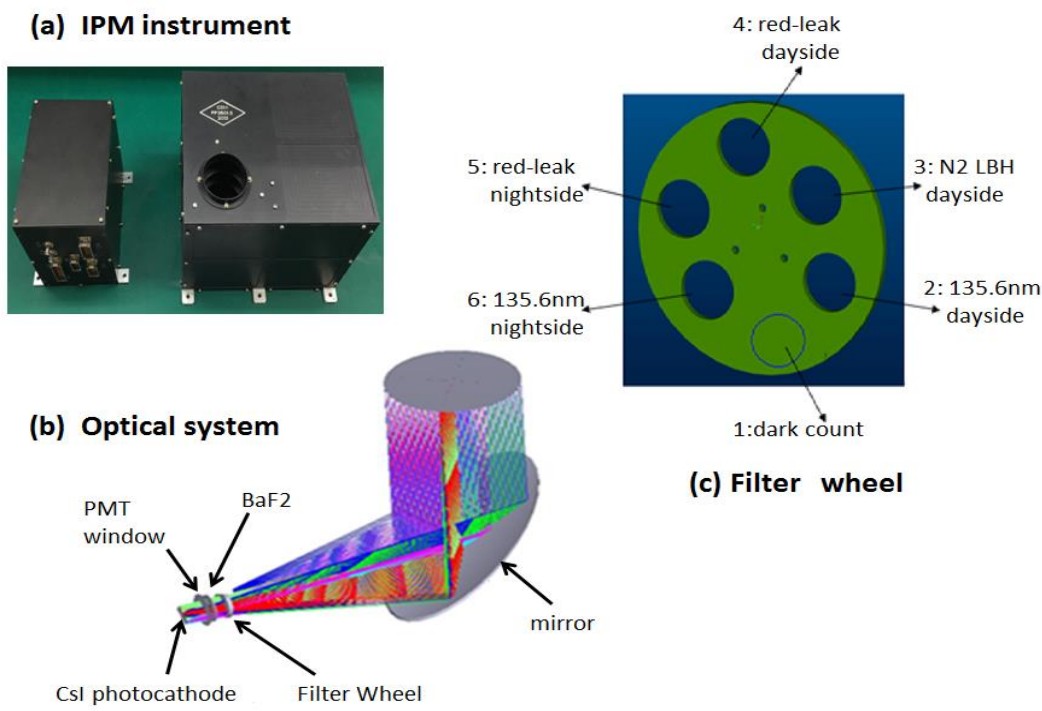






**Figure 1: IPM instrument.**

The IPM instrument is shown in Fig. 1 and includes a telescope, a filter wheel, a detector system, and control electronics cabinet. The telescope has a field-of-view of 3.5 ° (along orbit)×1.6 ° (cross orbit). An off-axis aluminum mirror coating $MgF_2$ is used to collected airglow emission in the telescope. To suppress the longer wavelength radiance, a sunblind PMT (R10825, Hamamatsu) with CsI photocathode is used in the detector system (Fu et al, 2015). The quantum efficiency of the PMT with an effective area of 4 ×9.5 mm, is about 26 % at the wavelength 135.6 nm, $6.17\times10^{-5}$ at 254 nm, and $4.06\times10^{-8}$ at 514 nm. The PMT has better than $10^{-4}$ rejection at wavelengths longer than 200 nm.

IPM monitors 135.6 nm emissions in the nighttime and 135.6 nm and $N_2$LBH emissions in the daytime by employing a filter wheel. There are six spots in the filter wheel (Fig. 1 (c)) corresponding to six channels of IPM: dark count channel, 135.6 nm nightside channel, red-leak nightside channel, red-leak dayside channel, $N_2$LBH dayside channel, and 135.6 nm dayside channel. The Channel information of IPM is shown in Table 2. In order to suppress the longer wavelength radiance further, the band-pass filter centred on 135.6 nm is used in the 135.6 nm dayside channel, and the band-pass filter centred on 160 nm is used in the $N_2$LBH channel. Besides, IPM specifically adds two red-leak signal channels for daytime and nighttime red-leak respectively. Based on the design of dayside or nightside channel, a $SiO_2$ filter is added in red-leak channels in order to eliminate below 180 nm wavelength airglow. By differencing the measurements of dayglow channels and red-leak dayside channel, dayglow radiations can be detected. And by differencing the measurements of 135.6 nm nightside channel and red-leak nightside channel, 135.6 nm radiation in the nighttime can be detected. To exclude radiation shorter than 135.6 nm completely, a 0.5 mm-thin VUV-grade BaF2 flat filter is used and the transmittance at 135.6 nm at room temperature is 0.5 (Fu et al., 2015). The emission of wavelengths shorter than 132 nm cannot pass the 0.5 mm-thick $BaF_2$ filter over a temperature range of 5 ℃ to 35 ℃.

**Table2. Channel information.**

| Number | Name | Filter |
|--------|------|--------|
| 1 | dark count channel | none |
| 2 | 135.6nm dayside channel | BaF2+bandpass |
| 3 | N2LBH dayside channel | BaF2+bandpass |
| 4 | red-leak dayside channel | BaF2+bandpass+quartz |
| 5 | red-leak nightside channel | BaF2+quartz |
| 6 | 135.6nm nightside channel | BaF2 |

IPM has two observation modes: day mode and night mode. The day mode includes 4 times observation of the 135.6 nm dayside channel, 4 times observation of the $N_2$LBH channel, 2 times observation of the red-leak dayside channel, and 1 dark



count observation in each frame. The night mode includes 8 times observation of the 135.6 nm night channel, 1 observation of the red-leak nightside channel, and 1 dark count observation.

**2.3 Laboratory Calibration**

The IPM was calibrated in ground laboratory prior to flight. The optical calibration facility in the ground laboratory has a deuterium lamp, a monochromator, a collimator, a diffuser board, and a NIST standard detector assembled in a modular pattern. The deuterium lamp (L11798) with a $MgF_2$ window has 150W power and provides a bright, stable source of FUV radiation. The source of FUV radiation is wavelength-selected by the monochromator (234/302) which has a $f$/4.5 0.2 m
Czerny-Turner with a 1200 grooves/mm grating. A collimator ensures that the beam consists of parallel rays. The NIST standard detector (AXUV-100G) provides a reference for calibrating IPM. The entire facility is installed in a vacuum environment which allows the propagation of radiation in the far ultraviolet.

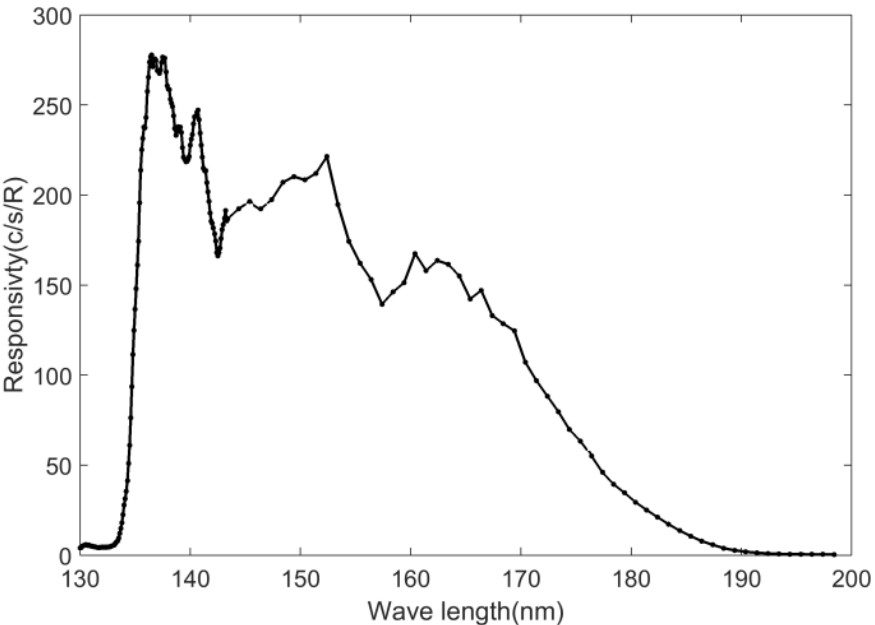

**Figure 2: The IPM responsivity of the 135.6nm nightside channel in counts/s/R.**


The processes of calibration are following: First, the FUV light at 125-200 nm from a deuterium lamp is selected by the monochromator. Second, the wavelength-selected light reaches the NIST standard detector through the collimator, and the NIST standard detector obtains the irradiance of the wavelength-selected light. And then, by using a rotating platform, the wavelength-selected light reaches the diffuser board through the collimator and enters IPM. IPM obtains the count of the
wavelength-selected light. Finally, the count and irradiance of the wavelength-selected light are used in calculating the responsivity to the wavelength-selected light. The uncertainty of the ground calibration comes from the stability of the FUV



light source, the error of the standard detector, the bi-directional reflection distribution function (BRDF) uncertainty of the diffuser board, the non-uniformity of the light source, and so on. The uncertainty of the ground calibration is estimated to reach 11.25%. As a function of wavelength, the responsivity of the 135.6 nm nightside channel from 130 to 200 nm is shown

in Figure 2. The responsivity to 135.6 nm radiation at night is about 266.9 counts/s/R near the peak of the responsivity function distribution, and reaches the design requirement of the 135.6 nm nightside channel. The responsivity to 135.6 nm radiation at night provides high sensitivity in observations of OI 135.6 nm radiation at night.

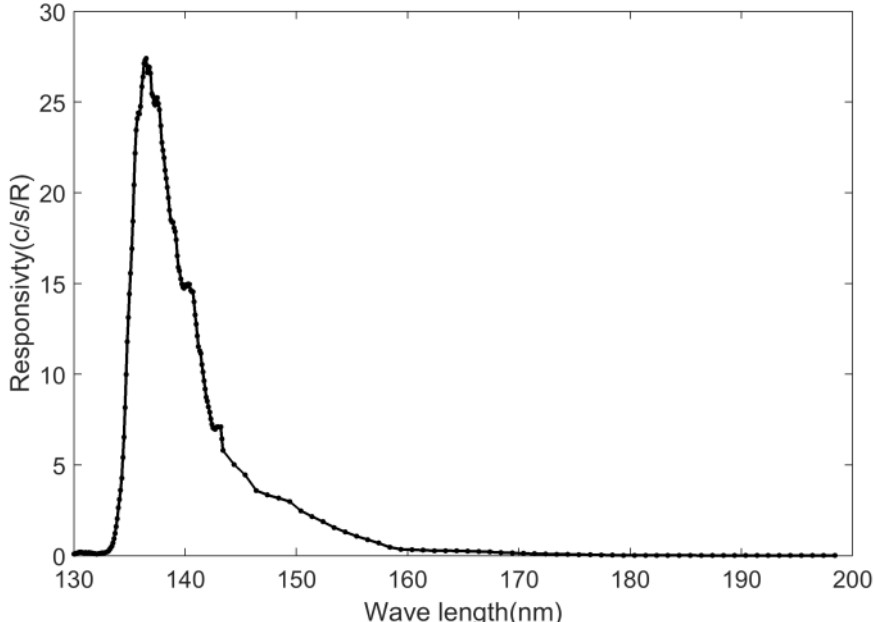

120              **Figure 3: The IPM responsivity of the 135.6nm dayside channel in counts/s/R.**

As a function of wavelength, the responsivity of the 135.6 nm dayside channel from 130 nm to 200 nm is shown in Figure 3. The responsivity to the 135.6 nm radiation in daytime is about 23.20 counts/s/R, and also reaches the design requirement of the 135.6 nm dayside channel. The responsivity is much less than the one on the nightside due to the bandpass used in the

135.6 nm dayside channel, which is designed to obtain the radiation of 135.6 nm in daytime and suppress the radiation at wavelengths shorter than 135.6 nm, $N_2LBH$ and red-leak in daytime. The other bandpass is used in the $N_2LBH$ day channel in order to obtain the radiation of $N_2LBH$ and suppress the radiation of 135.6 nm and red-leak in daytime. The responsivity of $N_2LBH$ channel is shown in Figure 4.



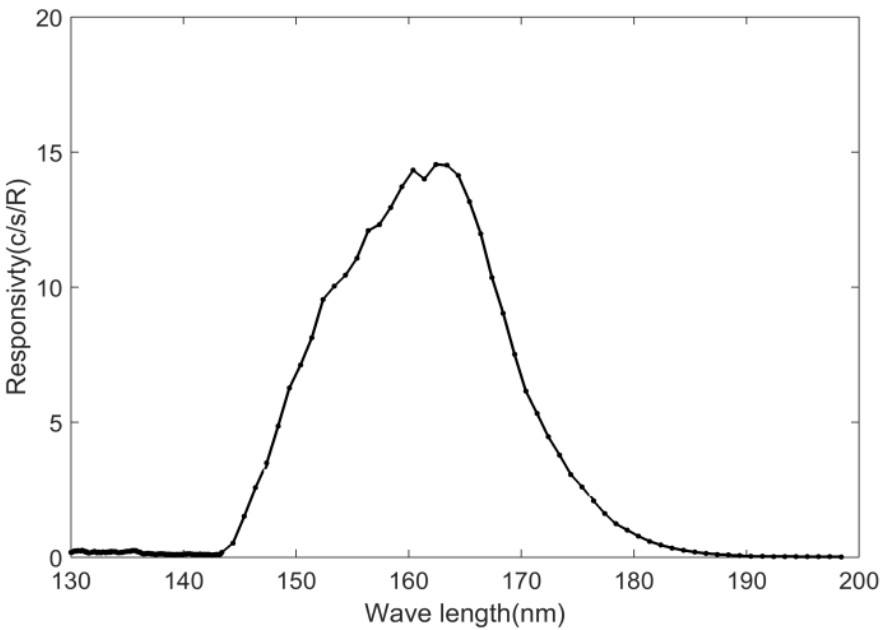

Figure 4: **The IPM responsivity of the N$_2$LBH channel in counts/s/R.**

## 3 Observation Results

### 3.1 OI 135.6 nm emission on the nightside

After the FY3D satellite was launched at 18:35 UTC on November 14, 2017, IPM started operation at 10:20 UTC on November 25, 2017. In IPM data processing, dark count is used to confirm the working status of IPM. Generally, the dark count of IPM is less than 10 counts. When the FY3D satellite passes by the South Atlantic Anomaly (SAA), the dark count of IPM increases rapidly and reaches a peak about 2000 counts due to the high energetic particles over the SAA.



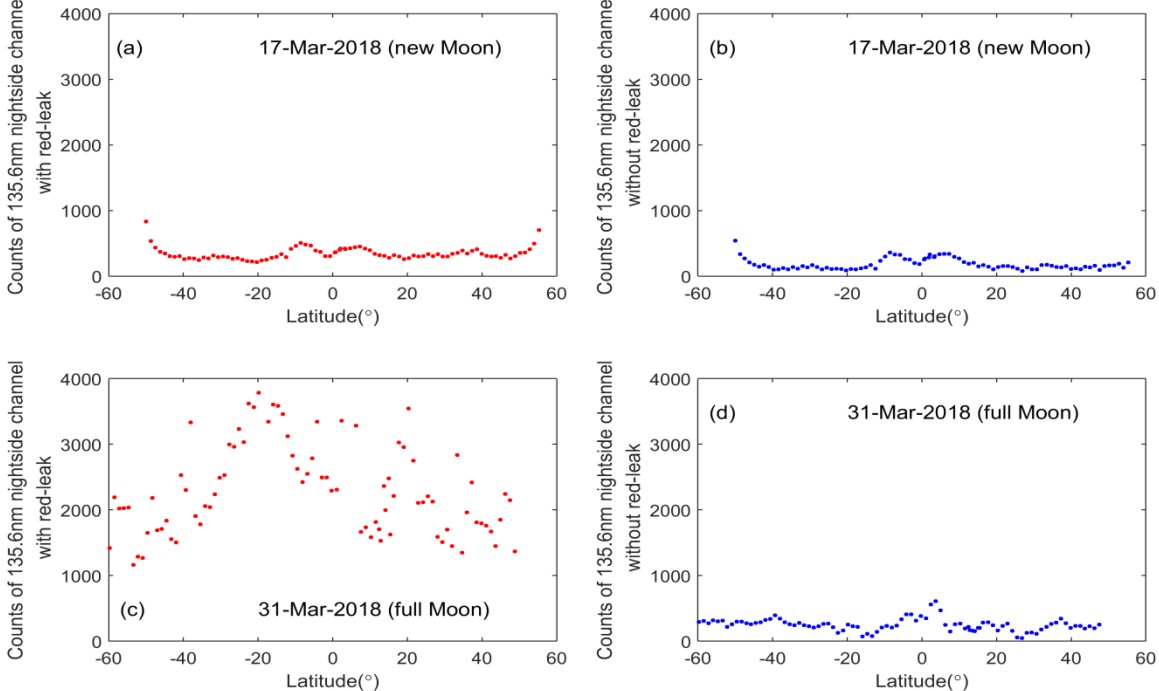

**Figure 5: The count of the 135.6nm nightside channel with (left) and without (right) red-leak for new Moon (top) and full Moon (bottom) situation, respectively. March 17, 2018 is new Moon day, and March 31, 2018 is full Moon day.**

The count of the 135.6 nm nightside channel is presented in Fig. 5. The count with red-leak on March 17, 2018 (new Moon) and on March 31, 2018 (full Moon) are shown in (a) and (c), respectively. The count without red-leak on March 17, 2018 and March 3, 2018 are shown in (b) and (d), respectively, which deducted the count of red-leak. The count of the 135.6 nm nightside channel in (c) is several times the count of the 135.6 nm nightside channel in (a) due to moonlight reflecting into the 135.6 nm nightside channel from cloud tops, while the count levels in (b) and (d) are very similar. We found that the red-leak nightside channel is effective to deduct the contamination of moonlight on the 135.6 nm nightside channel.

The example of the global count of the 135.6 nm nightside channel is presented in Fig. 6 (a). The red solid line indicates the magnetic dip equator. The data in Fig. 6 are from 7 to 11 December 2017. From 7 to 11 December 2017, Kp index is not more than 4 and the geomagnetic condition kept quiet relatively. As shown in Fig. 6 (a), there is a high-count area near the magnetic dip equator in South America, which shows the contamination in SAA associated with particles impacting the instrument. The example of global brightness of the 135.6 nm nightside channel without red-leak and the effect of dark count is presented in Fig. 6 (b). As shown Fig. 6 (b), there are some brighter areas located oneither side of the magnetic dip equator in South America and Africa, which are the so-called equatorial ionization anomaly (EIA) structure. EIA has been



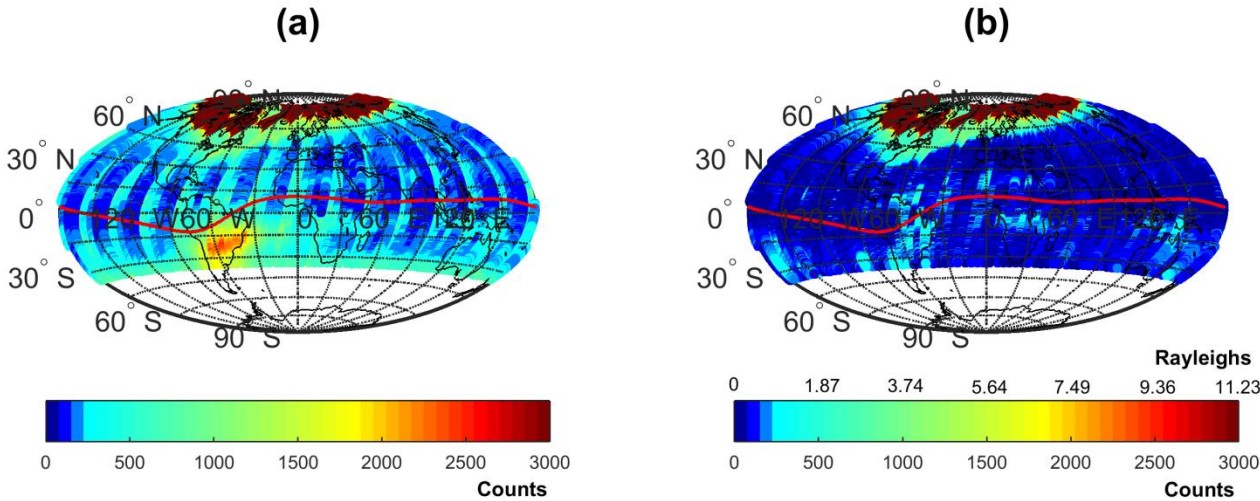

**Figure 6: The global count (left) and brightness (right) of the 135.6nm nightside channel from 7 to 11 December 2017. The brightness is without red-leak and the effect of dark count. The red solid line indicates the magnetic dip equator.**


studied extensively by using data from ground-based ionosodes (Moffett and Hanson, 1965; Walker, 1981) and ground-based optical observations (Thuillier et al., 1976). The OI 135.6 nm emission data from GUVI on board TIMED satellite, FUV on board the IMAGE satellite, and the TIP on board the COSMIC satellites have been used in study of the EIA phenomenon (Christensen et al., 2003; Sagawa et al., 2005; Immel et al, 2006 and Coker et al., 2009). The local time of the

IPM orbit on the nightside is 2:00 am. The EIA structure which we found at the 2:00 local time is later than other results mentioned earlier, and it need to be studied furtherly.

### 3.2 $NmF_2$ and TEC

OI 135.6 nm emission is one of the strongest lines in the FUV nightglow at low latitudes and has relatively high transparency in the upper atmosphere. In the nightside ionosphere, there are two primary production mechanisms of OI 135.6 nm emission:

(1) Atomic oxygen is excited through the recombination of atomic oxygen ions with electrons and produces OI 135.6 nm emission; (2) Atomic oxygen is excited through the mutual neutralization of $O^+$ with $O^-$ and produces OI 135.6 nm emission (Meier, 1991). The mutual neutralization has a relatively smaller contribution. The brightness of OI 135.6 nm emission varies with the electron density and the oxygen ion concentration basically. Equivalently, OI 135.6 nm emission is approximately proportional to the square of the electron density in the F-region.

Based on the previous studies of the nighttime OI 135.6 nm airglow using the radiative and emissive model, IRl2000 model, and MSISE90 model, the retrieval algorithm of $NmF_2$ derived from nighttime OI 135.6 nm emission was presented by Jiang et al. (2018). The brightness of the nighttime OI 135.6 nm emission is used to calculate ionospheric $NmF_2$ by the ratio

between NmF$_2$ and OI 135.6 nm emission from the retrieval algorithm. We selected the IPM derived NmF$_2$ data which were near to four IGGCAS ionosonde stations(Sanya (18.3 °N,109.6 °E), Wuhan (30.5 °N,114.4 °E), Beijing (40.3 °N,116.2 °E),

and Mohe (50.2 °N,122.5 °E)) from November 25, 2017 to May 8, 2018. Their difference in longitude were less than 12 ° and in latitude were less than 5 °. There is a standard deviation of 26.67% between IPM NmF$_2$ and IGGCAS ionosonde NmF$_2$ (shown in Fig. 7).

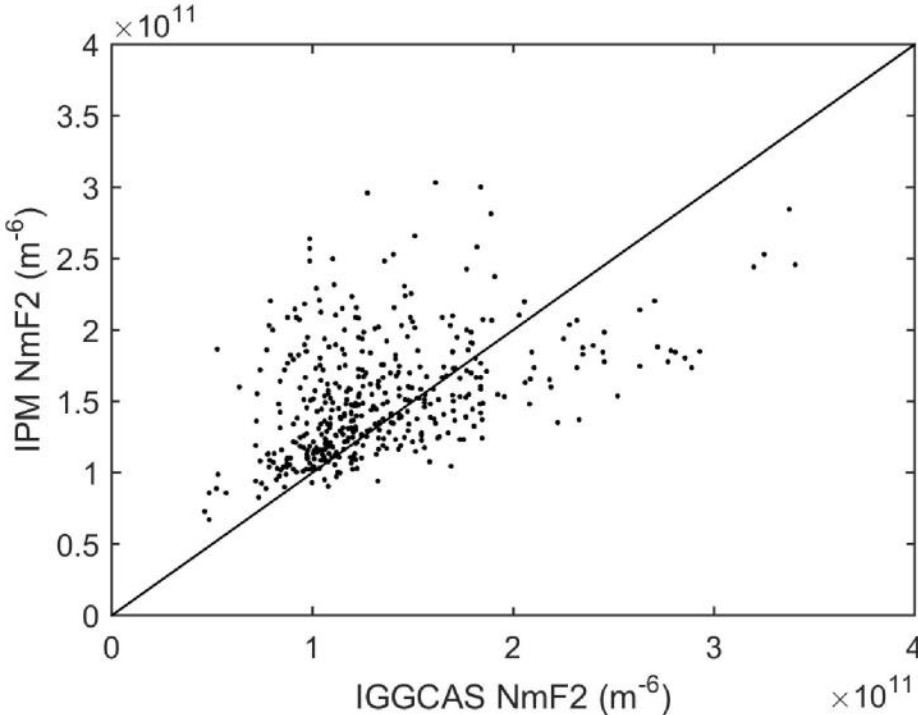

**Figure 7: IPM derived NmF2 and IGGCAS ionosondes NmF2 from November 25, 2017 to May 8, 2018. (The longitude difference between the IPM substellar point and ionosonde stations is less than 12 °, and the latitude difference is less than 5 °. )**

2014). We further calculated total electron content (TEC) from IPM results and compared with that of MIT TEC data from November 25, 2017 to April 8, 2018. The MIT TEC data (Rideout and Coster, 2006) was obtained from the MIT Haystack

Observatory Madrigal database (http://www.openmadrigal.org). There is a standard deviation of 39.41% between IPM TEC (total electron content unit, TECu) and MIT TEC (TECu) (shown in Fig.8). The standard deviation between IPM TEC (TECu) and MIT TEC (TECu) is more than the one between IPM NmF$_2$ and IGGCAS ionosonde NmF$_2$. In the Ionosphere plasmasphere coupled system, the ionosphere in conjugate hemispheres forms a plasmasphere reservoir along the interconnecting flux tube. There is diurnal interchange between the ionosphere and the plasmasphere that the downward

diffusion from the plasmasphere helps to maintain the nighttime F$_2$-layer. The results of Jason-1, Metop-A, and TerraSAR-X





(Yizengawa et al., 2008; Zakharenkova and Cherniak, 2015; Klimenko et al., 2015) show that at day the contribution of the plasmasphere in TEC is less than the one of the ionosphere, whereas at night the contribution of the plasmasphere in TEC is increasing and even more than the one of the ionosphere.

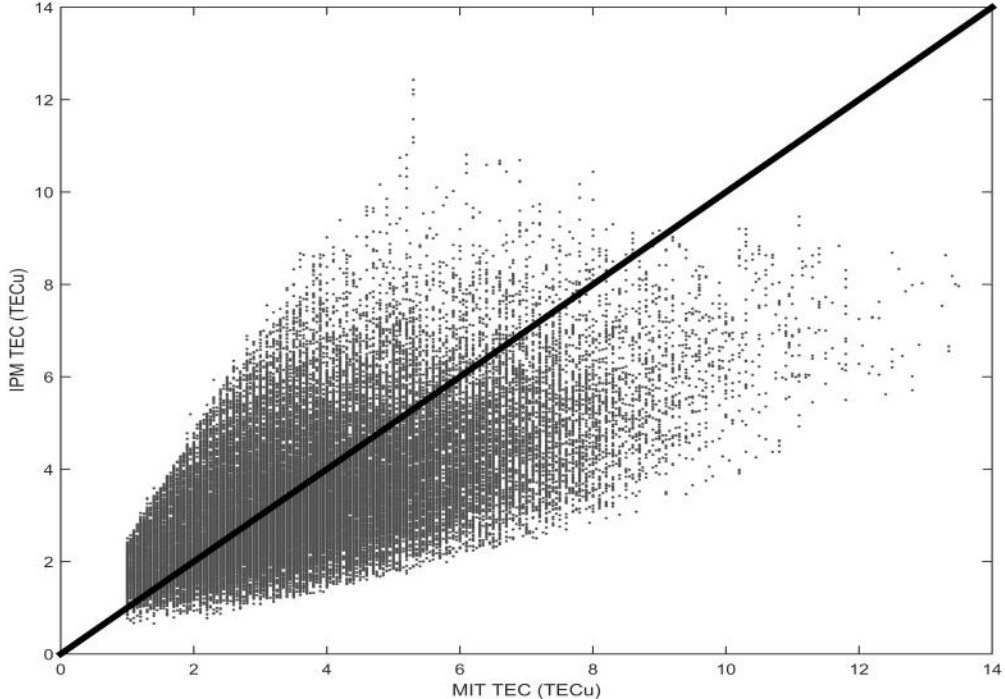

200                          **Figure 8: IPM TEC and MIT TEC (TECu) from November 25, 2017 to April 8, 2018**

Auroral emission can be derived from the 135.6 nm nightside channel. There is obviously a strong auroral emission feature in the Northern Hemisphere in Fig. 6. By the way, the wide-field auroral imager (WAI), one of ten scientific instruments aboard the Feng Yun 3D meteorological satellite, has provided large field of view (FOV), high spatial resolution, and
broadband ultraviolet images of the aurora (Zhang et al, 2019).



### 3.3 O/N$_2$

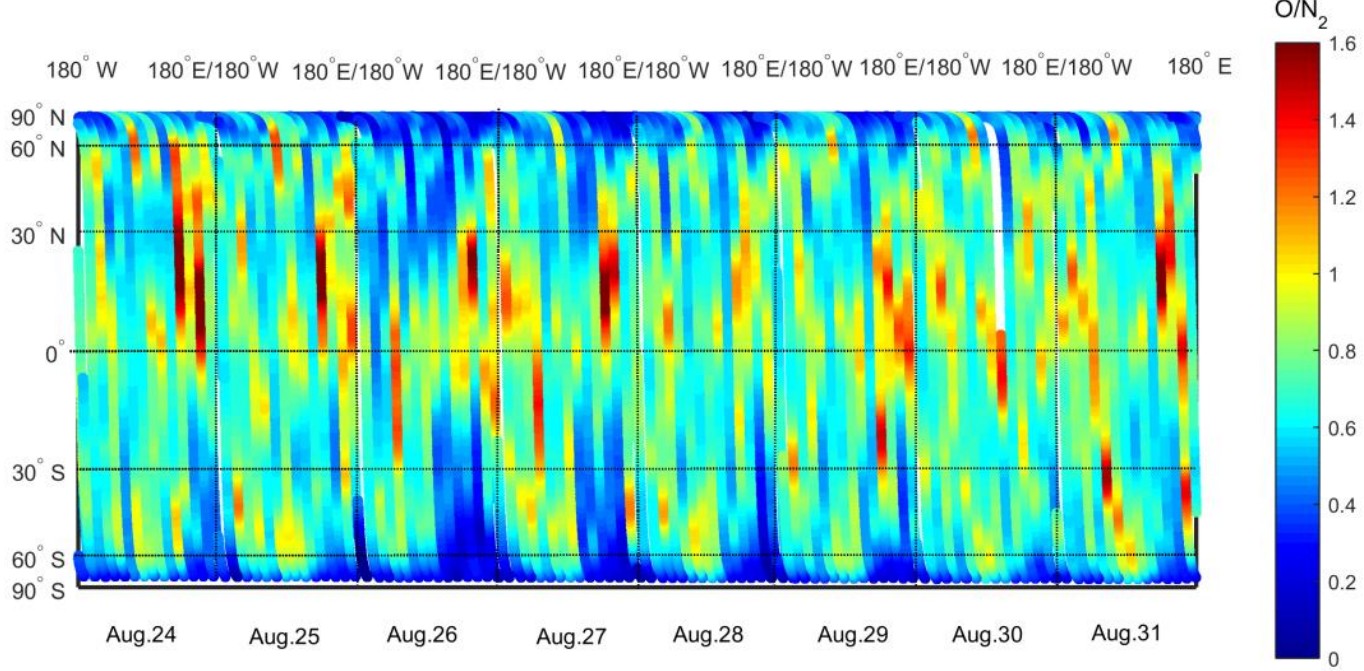

Figure 9:  Column O/N2 around the magnetic storm of Aug.26, 2018.

Energetic photon-electron impact excitation of the neutral atmosphere produces 135.6 nm emission and N$_2$LBH emission, which are proportional to the concentration of O and N$_2$ respectively (Meier, 1991).  135.6 nm emission and N$_2$LBH emission can be used to derive column O/N$_2$. The derivation of O/N$_2$ from disk 135.6 and N$_2$LBH dayglow observations was first addressed by Strickland et al. (Strickland et al., 1995) And the topic of O/N$_2$ from 135.6 nm emission and N$_2$LBH emission has been studied extensively (Christensen et al., 2014; Strickland et al., 2004; Zhang et al., 2014). During

geomagnetic storms enhanced Joule and particle heating in the high latitude ionosphere produces upwelling of the oxygen-depleted or nitrogen-rich air. The upwelling rises from much lower in the thermosphere into the F region. The heating also leads to enhanced horizontal equator-ward neutral winds that can change the distribution of the nitrogen-rich/oxygen-depleted air.

The brightness of 135.6 nm emission on dayside can be derived from observations of the 135.6 nm dayside channel and the

N$_2$LBH dayside channel respectively. A Butterworth filter is used in data processing in order to deduct the red-leak from the cloud top. The simulation produced by the AURIC model (Wang and Wang, 2015) is used to obtain a lookup table that provides the coefficient of deriving O/N$_2$ from a measured pair of 135.6 and LBHS values. The result of column O and N$_2$ ratio during the magnetic storm of Aug. 26, 2018 is presented in Fig. 9. On 24 August 2018 and most of 25 August 2018, Kp index is not more than 3. It abruptly rises 7 in 26 August 2018, descends and rises to 6 on 27 August 2018. From 29 to 31

August 2018, Kp index is not more than 3. The column $O/N_2$ on 24 and 25 August is relatively quiet, and significant changes in column $O/N_2$ occur on 26 and 27 August. The reduction of $O/N_2$ extends from the high-latitude region to mid and low latitude regions in the Northern and Southern Hemisphere. On 30 and 31 August, column $O/N_2$ returns to the level before the magnetic storm.

**4 Conclusion**

The FY3D satellite was launched at 18:35 UTC on November 14, 2017 from the Taiyuan Satellite Base, Shanxi province, China. The Ionospheric Photometer instrument carried aboard the Feng Yun 3D meteorological satellite measures the spectral radiance of the Earth far ultraviolet airglow in the spectral region from 133 to 180 nm. IPM is a tiny, highly sensitive, and robust remote sensing instrument. Preliminary observations show that the IPM could monitor the global structure of the equatorial ionization anomaly structure around 2:00 local time using OI 135.6 nm nightglow properly.  It could also identify

the reduction of $O/N_2$ in the high-latitude region during the geomagnetic storm of Aug. 26, 2018. The IPM derived $NmF_2$ accords well with that observed by 4 ionosonde stations along 120°E with a standard deviation of 26.67%. Initial results demonstrate that the performance of IPM meets the designed requirement and therefore can be used to study the thermosphere and ionosphere in future.

*Data availability.* Data are available at http://satellite.nsmc.org.cn/PortalSite/Default.aspx.

*Author contribution*s.  Yungang Wang and Tian Mao performed the data validation and prepared the paper and most of the plots; Liping Fu and Fang Jiang designed IPM and provided laboratory calibration data; Xiuqing Hu, Chengbao Liu, Xiaoxin Zhang, Jiawei Li, Ling Sun, Zhongdong Yang, Peng Zhang and Jingsong Wang participated in instrument parameters

requirements, judging of instrument design and data validation; Zhipeng Ren, Fei He and Lingfeng Sun participated in validation and intercomparisons.

*Competing interests.* The authors declare that they have no conflict of interest.

*Financial support.* This research has been supported by the Natural Science Foundation of China under Grant 41874187, 41774195, and 41931073 and Fengyun Satellite Ground Application System.

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
