# Peer review of "Far ultraviolet airglow remote sensing measurements on Feng Yun 3D meteorological satellite"

_Atmospheric Measurement Techniques, 2021_

## Author Comment (AC4)

**Response to Referee #1 :**

Response to the comments on amt-2021-195 by the reviewer 1.

General comments:

1. The most significant issue is that some information how the reported O/N2 values (and the brightness of the O and N2 emissions used in deriving the O/N2) compare with other observations, as was done form the NmF2 observations, is needed. The lack of such information is a major shortcoming of the current paper. Such information is needed for others to understand the value of the observations and their limitations.

**Answer: Thanks for the constructive comments. We added details on deriving the $O/N_2$ from the brightness of the O and $N_2$ emissions, and added the result of $O/N_2$ product compared with GUVI.**

**"Giving an $N_2$ depth of $10^{17}$ cm$^{-2}$, column O and $N_2$ ratio is derived from the value of at a given SZA by two dimensional interpolation. The retrieval algorithm could refer in relevant paper (Strickland et al., 1995; Zhang et al., 2004). The brightness of 135.6 nm emission and $N_2$ LBH emission on dayside were derived from observations of the 135.6 nm dayside channel and the $N_2$ LBH dayside channel respectively. In order to further deducting the red-leak from the cloud top, we used a Butterworth filter in data processing. The improved AURIC model (Wang and Wang, 2016) was used to produce a simulation. The simulation provided the coefficient of deriving $O/N_2$ from a measured pair of 135.6 and LBH. The column O and $N_2$ ratio during the magnetic storm of Aug. 26, 2018 was presented in Fig. 9a. On 24 August 2018 and most of 25 August 2018, Kp index was not more than 3. It abruptly rises to 7 in 26 August 2018. From 29 to 31 August 2018, Kp index was not more than 3. The column $O/N_2$ on 24 and 25 August was relatively quiet, and significant changes in column $O/N_2$ occurred on 26 and 27 August. The reduction of $O/N_2$ extended from the high-latitude region to mid- and low- latitude regions in the Northern and Southern Hemisphere. On 30 and 31 August, the column $O/N_2$ returned to quiet.**

**The column O and $N_2$ ratio derived from GUVI during the magnetic storm of Aug. 26, 2018 is presented in Fig. 9b.Fig. The GUVI column $O/N_2$ data (Strickland et al., 2004) was obtained from GUVI website (http://guvitimed.jhuapl.edu/data_fetch _l3_on2_ idlsave). The column $O/N_2$ from GUVI on 24 and 25 August was relatively quiet, and significant changes in column $O/N_2$ occurred on 26 and 27 August. The reduction of $O/N_2$ also extended from the high-latitude region to mid- and low- latitude regions in the Northern and Southern Hemisphere. On 30 and 31 August, the column $O/N_2$ of GUVI also returned to quiet. The features of column $O/N_2$ of IPM and GUVI during the magnetic storm of Aug. 26, 2018 were similar. These results showed that the IPM data could provide a good monitoring of $O/N_2$ changes during the magnetic storm."**

[Figure]

**Figure 9a: Column O/N2 from IPM around the magnetic storm of Aug.26, 2018.**

[Figure]

**Figure 9b: Column O/N2 from GUVI around the magnetic storm of Aug.26, 2018.**

2. A minor issue is that substantial editing for clarity is also needed the paper. Some specific suggestions are included below.

**Answer:** **We re-edited the whole paper according to the referee's comments.**

The observations described are potentially very valuable for understanding the effects of geomagnetic activity on the composition of the thermosphere and worthy of publication if their relative differences from and consistency with other O/N2 observations can be quantified.

3. Specific comments and technical corrections:

- Line 19: delete "properly".

  **[It is corrected according to the referee's comments.]**

- Line 21: change "designed requirement" to "design requirements"

**[It is corrected according to the referee's comments.]**

- Line 26: "can present" doesn't work well, perhaps "represents the" or "is representative of the"?

**[It is corrected according to the referee's comments.]**

- Line 27: add space between "N$_2$" and "LBH" (also throughout the paper) and "can be" to "is". Wording of the sentences could be more concise ("FUV radiation" is repeated).

**[We added a space between "N$_2$" and "LBH" throughout the manuscript. We have reworded the sentence: "The Earth's atmosphere is opaque to the FUV radiation due to the lower atmosphere absorption.". ]**

- Line 28: delete "characteristics of"

**[It is corrected according to the referee's comments.]**

- Line 29: Delete the sentence that begins on this line or reword to combine with previous sentence. It's redundant with the previous sentence.

**[We have reworded the sentence: "The Earth's atmosphere is opaque to the FUV radiation due to the lower atmosphere absorption.".]**

- Line 31-33: move "based on satellites" to after "ionosphere" and reword slightly ("from satellites" perhaps). Also, delete comma after "as" and "2003)"

**[We have reworded the sentence: "In past decades, FUV spectrography has been used extensively in studying the thermosphere and ionosphere from satellites, such as GUVI (the Global Ultra-Violet Imager) on the NASA TIMED (Thermosphere, Ionosphere, Mesosphere Energetics and Dynamics) satellite (Christensen et al., 2003) and the Far Ultraviolet Imager (FUV) on the NASA IMAGE (Imager for Magnetopause-to-Aurora Global Exploration) satellite (Sagawa 35 et al., 2005).".]**

- Line 35: If the photometer is being used for the thermosphere also, why is it an "ionospheric photometer"? change "equipment" to "instrument".

**[Good suggestion! But the instrument has been named and we can't change it. Maybe we can use this name for the next generation.]**

**[We have reworded the sentence: "The other useful instrument is ionospheric photometer, which is compact and high-sensitive."]**

- Line 36: Photometer suitable for observations of the nighttime ionosphere were made and flown decades before NRL's instrument (e.g., https://doi.org/10.1029/JA085iA05p02201, and there are probably earlier examples).

**[Thanks. We added "The photometer on the polar-orbiting Department of Defense satellite S3-4 was used in measuring of the airglow, aurora, and solar scatter radiance of the earth's atmosphere (Huffman et al., 1980).".]**

**[We deleted "firstly".]**

- Line 46: "leak" to "leaks"

  **[It is corrected according to the referee's comments.]**

- Line 52: suggest changing "and designed" to "and is designed"

  **[It is corrected according to the referee's comments.]**

- Line 57: delete "parameters"

  **[It is corrected according to the referee's comments.]**

- Line 59: use plural "photometers"

  **[It is corrected according to the referee's comments.]**

- Line 81: "added in" to "added in the"

  **[It is corrected according to the referee's comments.]**

- Line 82: Rewording is needed. Airglow at longer wavelengths is the problem, the wording "airglow below 180 nm" indicates that it is the shorter wavelengths. May be clearer to say whether it is longer or shorter wavelengths.

  **[We have reworded the sentence: "Based on the design of dayside or nightside channel, a SiO$_2$ filter is added in red-leak channels in order to eliminate emission longer than 180 nm.".]**

- Lines 91-93: delete "times" and use plural "observations" for cases with >1 observation.

  **[It is corrected according to the referee's comments.]**

- Line 96: first "ground" to "the" and delete "in the ground laboratory".

  **[It is corrected according to the referee's comments.]**

- Line 101-102: The deuterium lamp is completely with a vacuum environment, and the monochromator? That's what the wording indicates. Seems unlikely and at least unnecessary. A more accurate description may be needed. Typical facility would have vacuum only within the monochromator.

  **[We have reworded the paragraph: "The optical calibration facility in ground has a deuterium lamp, a monochromator, a collimator, a diffuser, a standard detector and a vacuum chamber assembled in a modular pattern(Fig. 2). The deuterium lamp (L11798) with a MgF$_2$ window has 150W power and provides a bright, stable source of**

**FUV radiation. The source of FUV radiation is wavelength-selected by the monochromator (234/302) which has a ƒ/4.5 0.2 m 100 Czerny-Turner with a 1200 grooves/mm grating. A collimator ensures that the beam consists of parallel rays. The standard detector (AXUV-100G) traced from NIST provides a reference for calibrating IPM."]**

[Figure]

Figure 2: The optical calibration facility in ground

**[We deleted the sentence "The entire facility is installed in a vacuum environment which allows the propagation of radiation in the far ultraviolet.".]**

- Line 106: delete "following" and "from a" to "from the" (unless multiple lamps are attached at the same time).

  **[It is corrected according to the referee's comments.]**

- Lines 107, 108, 109, 110, 111: "wavelength-selected light" to "wavelength selected"?

  **[Yes, It is corrected according to the referee's comments.]**

- Line 109: "counts", plural, or "signal"; possible "for" rather than "of".      ????

  **[Yes, It is corrected according to the referee's comments.]**

- line 126: "red-leak in daytime" to "red-leak contributions in the daytime"?

  **[Yes, It is corrected according to the referee's comments.]**

- line 127: "red-leak in daytime" to "red-leak contributions"?

  **[Yes, It is corrected according to the referee's comments.]**

- Line 137: < 10 per second, or what time interval?

  **[Yes, It is corrected according to the referee's comments.]**

- Line 138: delete "a peak" and "high".

  **[It is corrected according to the referee's comments.]**

- 2000 counts for what time interval?

  **["2000 counts" should be "2000 counts per second". It is corrected according to the referee's comments.]**

- Might also substitute "in" for "over" since the satellite is flying through region containing energetic particles.

  **[It is corrected according to the referee's comments.]**

- Line 145: by "without red-leak" you mean "with the red-leak signal subtracted"?

  **[Yes.]**

- Line 146: delete "which deducted the count of red-leak"? (phrase seems redundant)

  **[Yes, It is corrected according to the referee's comments.]**

- Line 149: "deduct"?    ???

  **[We have replaced "deduct" with "eliminate".]**

- Line 150: "The" to "An"

  **[It is corrected according to the referee's comments.]**

- Line 153: "condition kept quiet relatively" to "conditions were relatively quiet"

  **[It is corrected according to the referee's comments.]**

- Line 154: "The example" to "An example", unless these are the only data collected.

  **[It is corrected according to the referee's comments.]**

- Line 155: "oneither" needs a space between words.

  **[It is corrected according to the referee's comments.]**

- Line 156: "EIA has" to "The EIA has"?

  **[Yes, it is corrected according to the referee's comments.]**

- Line 163: "have been" to "have also been"?

  **[Yes, it is corrected according to the referee's comments.]**

- Line 166: "further" rather than "furtherly"

  **[It is corrected according to the referee's comments.]**

- Line 187: "2014)."? Something seems to be missing.

  **[Yes, one sentence was missing. We added "Similarly, the brightness of the nighttime OI 135.6 nm emission is also used to calculate ionospheric TEC by the ratio between TEC and the nighttime OI 135.6 nm emission (Jiang et al., 2014).".]**

- Line 181-190: is this agreement you give with NmF2 and TEC typical when observations are compared?

  **[Yes, it is.]**

- Lines 194-195: wording in the phrase "that the…F2-layer" doesn't make complete sense, it almost does but something seems off.

  **[We have reworded the sentence: "There is diurnal interchange between the ionosphere and the plasmasphere, the downward diffusion from the plasmasphere helps to maintain the nighttime $F_2$-layer."]**

- Line 196-198: wording, "at day the contribution…of the ionosphere." Needs some rewording to be clear (unclear what "less than one of the ionosphere" means) and somewhat odd use of prepositions ("at day the contribution" to "on the dayside, contributions" perhaps).

  **["The results of Jason-1, Metop-A, and TerraSAR-X (Yizengawa et al., 2008; Zakharenkova and Cherniak, 2015; Klimenko et al., 2015) show the plasmasphere contribution at night can't be neglected."]**

  **["MIT TEC is intergraded from ground to 20200Km. It includes plasmasphere contribution and ionosphere contribution. IPM TEC is intergraded from ground to 830Km, it only includes ionosphere contribution."]**

- Line 201: "Auroral emission can be derived from the 135.6…"? But 135.6 nm is an auroral emission, nothing to derive, just observe it.

  **[We deleted the paragraph about aurora after considerations.]**

- Lines 203-205: Unclear why the sentence that begins on line 203 is included. This is the only mention of the WAI instrument in the paper. If it is included, more relevance to the current paper should be added.

  **[We deleted the paragraph about aurora after considerations.]**

- Lines 210-213: The two sentences, immediately following the auroral discussion, may give an incorrect impression. Column O/N2 derivation would be from dayglow observations, but not from auroral.

  **[We deleted the paragraph about aurora after considerations.]**

- Line 219-220: Needs some minor rewording for clarity, unless the 135.6 nm brightness can be derived from either channel. Maybe just say the 135.6 nm brightness can be derived "using" the dayside 135.6 nm and N2 LBH channels?

**[Sorry, we missed some words. We added "The brightness of 135.6 nm emission and $N_2$ LBH emission on dayside can be derived from observations of the 135.6 nm dayside channel and the $N_2$ LBH dayside channel respectively.".]**

- Line 221: "cloud top" to "cloud tops"

**[It is corrected according to the referee's comments.]**

- Line 223-224: "result of column O and $N_2$ ratio", simplify to "column O/$N_2$ ratio"?

**[It is corrected according to the referee's comments.]**

- Line 230: suggest spelling out the name of the satellite (FY3D) the first time it's mentioned in the conclusions, then using the abbreviation (as on line 231) if desired.

**[It is corrected according to the referee's comments.]**

- Line 235: How do the changes seen in O/N2 compare with other observations? There are probably TIMED/GUVI observations available to compare against. Later storms could possibly be compared with O/N2 observations from the GOLD mission, in addition to GUVI. Some comparison of the reported O/N2 values with other observations, as was done form the NmF2 observations, is needed in the paper.

[**We have added the result of O/$N_2$ product compared with GUVI. Please refer to the first item of our response.**]

---

## Author Response (AR1)

**Response to Referee #1 :**

Response to the comments on amt-2021-195 by the reviewer 1.

General comments:

1. The most significant issue is that some information how the reported O/N2 values (and the brightness of the O and N2 emissions used in deriving the O/N2) compare with other observations, as was done form the NmF2 observations, is needed. The lack of such information is a major shortcoming of the current paper. Such information is needed for others to understand the value of the observations and their limitations.

**Answer: Thanks for the constructive comments. (1) We added details on deriving the O/$N_2$ from the brightness of the O and $N_2$ emissions according to the referee's comments. See Line 229-234 in the revised manuscript. (2) And we also added the result of the O/$N_2$ from IPM compared with the O/$N_2$ from GUVI. See Line 235-246, Fig. 11 and Fig.12 in the revised manuscript.**

**"Giving an $N_2$ depth of $10^{17}$ cm$^{-2}$, column O and $N_2$ ratio is derived from the value of at a given Solar Zenith Angle (SZA) by two dimensional interpolation. The retrieval algorithm could refer in relevant paper (Strickland et al., 1995; Zhang et al., 2004). The brightness of 135.6 nm emission and $N_2$ LBH emission on dayside were derived from observations of the 135.6 nm dayside channel and the $N_2$ LBH dayside channel respectively. In order to further deducting the red-leak from the cloud top, we used a Butterworth filter in data processing. The improved AURIC model (Wang and Wang, 2016) was used to produce a simulation. The simulation provided the coefficient of deriving O/$N_2$ from a measured pair of 135.6 and LBH. The column O and $N_2$ ratio during the magnetic storm of Aug. 26, 2018 was presented in Fig. 11. On 24 August 2018 and most of 25 August 2018, Kp index was not more than 3. It abruptly rises to 7 in 26 August 2018. From 29 to 31 August 2018, Kp index was not more than 3. The column O/$N_2$ on 24 and 25 August was relatively quiet, and significant changes in column O/$N_2$ occurred on 26 and 27 August. The reduction of O/$N_2$ extended from the high-latitude region to mid- and low- latitude regions in the Northern and Southern Hemisphere. On 30 and 31 August, the column O/$N_2$ returned to quiet.**

**The column O and $N_2$ ratio derived from GUVI during the magnetic storm of Aug. 26, 2018 is presented in Fig. 12. The GUVI column O/$N_2$ data (Strickland et al., 2004) was obtained from GUVI website (http://guvitimed.jhuapl.edu/data_fetch _l3_on2_ idlsave). The column O/$N_2$ from GUVI on 24 and 25 August was relatively quiet, and significant changes in column O/$N_2$ occurred on 26 and 27 August. The reduction of O/$N_2$ also extended from the high-latitude region to mid- and low- latitude regions in the Northern and Southern Hemisphere. On 30 and 31 August, the column O/$N_2$ of GUVI also returned to quiet. The features of column O/$N_2$ of IPM and GUVI during the magnetic storm of Aug. 26, 2018 were similar. These results showed that the IPM data could provide a good monitoring of O/$N_2$ changes during the magnetic storm."**

[Figure]

**Figure 11: Column O/N₂ from IPM around the magnetic storm of Aug.26, 2018.**

[Figure]

**Figure 12: Column O/N₂ from GUVI around the magnetic storm of Aug.26, 2018.**

2. A minor issue is that substantial editing for clarity is also needed the paper. Some specific suggestions are included below.

**Answer:We re-edited the whole paper according to the referee's comments.**

The observations described are potentially very valuable for understanding the effects of geomagnetic activity on the composition of the thermosphere and worthy of publication if their relative differences from and consistency with other O/N2 observations can be quantified.

3. Specific comments and technical corrections:

- Line 19: delete "properly".

   **[It is corrected according to the referee's comments.]**

- Line 21: change "designed requirement" to "design requirements"

**[It is corrected according to the referee's comments. See Line 21 in the revised manuscript.]**

- Line 26: "can present" doesn't work well, perhaps "represents the" or "is representative of the"?

**[It is corrected according to the referee's comments. See Line 26 in the revised manuscript.]**

- Line 27: add space between "$N_2$" and "LBH" (also throughout the paper) and "can be" to "is". Wording of the sentences could be more concise ("FUV radiation" is repeated).

**[We added a space between "$N_2$" and "LBH" throughout the manuscript. We have reworded the sentence. See Line 28-29 in the revised manuscript.]**

- Line 28: delete "characteristics of"

**[It is corrected according to the referee's comments.]**

- Line 29: Delete the sentence that begins on this line or reword to combine with previous sentence. It's redundant with the previous sentence.

**[We have reworded the sentence according to the referee's comments. See Line 28-29 in the revised manuscript.]**

- Line 31-33: move "based on satellites" to after "ionosphere" and reword slightly ("from satellites" perhaps). Also, delete comma after "as" and "2003)"

**[They are corrected according to the referee's comments. See Line 31-33 in the revised manuscript.]**

- Line 35: If the photometer is being used for the thermosphere also, why is it an "ionospheric photometer"? change "equipment" to "instrument".

**[Good suggestion! But the instrument has been named and we can't change it. Maybe we can use this name for the next generation.]**

**[We have reworded the sentence according to the referee's comments. See Line 34-35 in the revised manuscript.]**

- Line 36: Photometer suitable for observations of the nighttime ionosphere were made and flown decades before NRL's instrument (e.g., https://doi.org/10.1029/JA085iA05p02201, and there are probably earlier examples).

**[Thanks. We added: "The photometer on the polar-orbiting Department of Defense satellite S3-4 was used in measuring of the airglow, aurora, and solar scatter radiance of the earth's atmosphere (Huffman et al., 1980)". See Line 35-36 in the revised manuscript.]**

- Line 46: "leak" to "leaks"

**[It is corrected according to the referee's comments. See Line 46 in the revised manuscript.]**

- Line 52: suggest changing "and designed" to "and is designed"

**[It is corrected according to the referee's comments. See Line 52 in the revised manuscript.]**

- Line 57: delete "parameters"

**[It is corrected according to the referee's comments.]**

- Line 59: use plural "photometers"

**[It is corrected according to the referee's comments.  See Line 59 in the revised manuscript.]**

- Line 81: "added in" to "added in the"

**[It is corrected according to the referee's comments. See Line 81 in the revised manuscript.]**

- Line 82: Rewording is needed. Airglow at longer wavelengths is the problem, the wording "airglow below 180 nm" indicates that it is the shorter wavelengths. May be clearer to say whether it is longer or shorter wavelengths.

**[We have reworded the sentence according to the referee's comments. See Line 81-82 in the revised manuscript.]**

- Lines 91-93: delete "times" and use plural "observations" for cases with >1 observation.

**[It is corrected according to the referee's comments. See Line 91-93 in the revised manuscript.]**

- Line 96: first "ground" to "the" and delete "in the ground laboratory".

**[It is corrected according to the referee's comments.]**

- Line 101-102: The deuterium lamp is completely with a vacuum environment, and the monochromator? That's what the wording indicates. Seems unlikely and at least unnecessary. A more accurate description may be needed. Typical facility would have vacuum only within the monochromator.

**[We have reworded the paragraph added the figure about optical calibration facility in ground according to the referee's comments. See Line 96-101 and Fig.2 in the revised manuscript.]**

**["The optical calibration facility in ground has a deuterium lamp, a monochromator, a collimator, a diffuser, a standard detector and a vacuum chamber assembled in a modular pattern(Fig. 2). The deuterium lamp (L11798) with a MgF$_2$ window has 150W**

**power and provides a bright, stable source of FUV radiation. The source of FUV radiation is wavelength-selected by the monochromator (234/302) which has a *f*/4.5 0.2 m 100 Czerny-Turner with a 1200 grooves/mm grating. A collimator ensures that the beam consists of parallel rays. The standard detector (AXUV-100G) traced from NIST provides a reference for calibrating IPM."]**

[Figure]

Figure 2: The optical calibration facility in ground

- Line 106: delete "following" and "from a" to "from the" (unless multiple lamps are attached at the same time).

  **[It is corrected according to the referee's comments. See Line 105 in the revised manuscript.]**

- Lines 107, 108, 109, 110, 111: "wavelength-selected light" to "wavelength selected"?

  **[Yes, they are corrected according to the referee's comments. See Line 106-110 in the revised manuscript. ]**

- Line 109: "counts", plural, or "signal"; possible "for" rather than "of".      ????

  **[Yes, they are corrected according to the referee's comments. See Line 108 in the revised manuscript.]**

- line 126: "red-leak in daytime" to "red-leak contributions in the daytime"?

  **[Yes, it is corrected according to the referee's comments. See Line 126 in the revised manuscript.]**

- line 127: "red-leak in daytime" to "red-leak contributions"?

**[Yes, it is corrected according to the referee's comments. See Line 128 in the revised manuscript.]**

- Line 137: < 10 per second, or what time interval?

  **[Yes, it is corrected according to the referee's comments. See Line 137 in the revised manuscript.]**

- Line 138: delete "a peak" and "high". 2000 counts for what time interval?

  **[They are corrected according to the referee's comments. See Line 138 in the revised manuscript.]**

  **["2000 counts" should be "2000 counts per second". It is corrected according to the referee's comments.]**

- Might also substitute "in" for "over" since the satellite is flying through region containing energetic particles.

  **[It is corrected according to the referee's comments. See Line 138 in the revised manuscript.]**

- Line 145: by "without red-leak" you mean "with the red-leak signal subtracted"?

  **[Yes.]**

- Line 146: delete "which deducted the count of red-leak"? (phrase seems redundant)

  **[Yes, it is corrected according to the referee's comments. See Line 147 in the revised manuscript.]**

- Line 149: "deduct"?    ???

  **[We have replaced "deduct" with "eliminate". See Line 150 in the revised manuscript.]**

- Line 150: "The" to "An"

  **[It is corrected according to the referee's comments. See Line 151 in the revised manuscript.]**

- Line 153: "condition kept quiet relatively" to "conditions were relatively quiet"

  **[It is corrected according to the referee's comments. See Line 153 in the revised manuscript.]**

- Line 154: "The example" to "An example", unless these are the only data collected.

  **[It is corrected according to the referee's comments. See Line 155 in the revised manuscript.]**

- Line 155: "oneither" needs a space between words.

**[It is corrected according to the referee's comments. See Line 151 in the revised manuscript.]**

- Line 156: "EIA has" to "The EIA has"?

**[Yes, it is corrected according to the referee's comments. See Line 157 in the revised manuscript.]**

- Line 163: "have been" to "have also been"?

**[Yes, it is corrected according to the referee's comments. See Line 164 in the revised manuscript.]**

- Line 166: "further" rather than "furtherly"

**[It is corrected according to the referee's comments.]**

- Line 187: "2014)."? Something seems to be missing.

**[Yes, one sentence was missing. We have added the sentence. See Line 193-194 in the revised manuscript.]**

- Line 181-190: is this agreement you give with NmF2 and TEC typical when observations are compared?

**[Yes, it is.]**

- Lines 194-195: wording in the phrase "that the…F2-layer" doesn't make complete sense, it almost does but something seems off.

**[We have reworded the sentence according to the referee's comments. See Line 201-203 in the revised manuscript.]**

- Line 196-198: wording, "at day the contribution…of the ionosphere." Needs some rewording to be clear (unclear what "less than one of the ionosphere" means) and somewhat odd use of prepositions ("at day the contribution" to "on the dayside, contributions" perhaps).

**[We have reworded the sentence according to the referee's comments. See Line 204-205 and Line 200-201 in the revised manuscript.]**

- Line 201: "Auroral emission can be derived from the 135.6…"? But 135.6 nm is an auroral emission, nothing to derive, just observe it.

**[We deleted the paragraph about aurora after considerations.]**

- Lines 203-205: Unclear why the sentence that begins on line 203 is included. This is the only mention of the WAI instrument in the paper. If it is included, more relevance to the current paper should be added.

**[We deleted the paragraph about aurora after considerations.]**

- Lines 210-213: The two sentences, immediately following the auroral discussion, may give an incorrect impression. Column O/N2 derivation would be from dayglow observations, but not from auroral.

**[We deleted the paragraph about aurora after considerations.]**

- Line 219-220: Needs some minor rewording for clarity, unless the 135.6 nm brightness can be derived from either channel. Maybe just say the 135.6 nm brightness can be derived "using" the dayside 135.6 nm and N2 LBH channels?

**[Sorry, we missed some words. We have reworded the sentence. See Line 230-232 in the revised manuscript.]**

- Line 221: "cloud top" to "cloud tops"

**[It is corrected according to the referee's comments. See Line 232 in the revised manuscript.]**

- Line 223-224: "result of column O and $N_2$ ratio", simplify to "column $O/N_2$ ratio"?

**[It is corrected according to the referee's comments. See Line 234 in the revised manuscript.]**

- Line 230: suggest spelling out the name of the satellite (FY3D) the first time it's mentioned in the conclusions, then using the abbreviation (as on line 231) if desired.

**[It is corrected according to the referee's comments. See Line 250-251 in the revised manuscript.]**

- Line 235: How do the changes seen in O/N2 compare with other observations? There are probably TIMED/GUVI observations available to compare against. Later storms could possibly be compared with O/N2 observations from the GOLD mission, in addition to GUVI. Some comparison of the reported O/N2 values with other observations, as was done form the NmF2 observations, is needed in the paper.

**[We also added the result of the O/N2 from IPM compared with the O/N2 from GUVI.. Please refer to the first item of our response.]**

**Response to Referee #2 :**

Response to the comments on amt-2021-195 by the Referee #2.

1. The manuscript describes the data and products from an ionospheric photometer (IPM). The IPM measures FUV emissions, such as O 135.6 nm and $N_2$ LBH bands. It is a sensitive instrument. The data products include TEC, $NmF_2$ and $O/N_2$ column density ratio. They are useful data products. However, it lacks of details on the methods for estimating these products. The $O/N_2$ product is not validated or compared with existing $O/N_2$ data from other missions. Some of the $O/N_2$ features, likely artifacts, are not discussed.

**Answer**:**Thanks for the constructive comments. (1) We added details on deriving the $O/N_2$ from the brightness of the O and $N_2$ emissions according to the referee's comments. See Line 229-234 in the revised manuscript. (2) And we also added the result of the $O/N_2$ from IPM compared with the $O/N_2$ from GUVI. See Line 235-246, Fig. 11 and Fig.12 in the revised manuscript. (3) The IPM monitors 135.6 nm and $N_2$ LBH emissions in the day-side thermosphere by employing a filter wheel. The 135.6 nm dayside channel and the $N_2$ LBH dayside channel are asynchronous during observation. Fast varies from cloud tops could enter these channels, and make the count of these channels increase sharply. It leads to some of the $O/N_2$ features, such as sporadic enhanced $O/N_2$. We used a Butterworth filter in order to eliminate fast varies from cloud tops.**

**["Giving an $N_2$ depth of $10^{17}$ $cm^{-2}$, column O and $N_2$ ratio is derived from the value of at a given Solar Zenith Angle (SZA) by two dimensional interpolation. The retrieval algorithm could refer in relevant paper (Strickland et al., 1995; Zhang et al., 2004). The brightness of 135.6 nm emission and $N_2$ LBH emission on dayside were derived from observations of the 135.6 nm dayside channel and the $N_2$ LBH dayside channel respectively. In order to further deducting the red-leak from the cloud top, we used a Butterworth filter in data processing. The improved AURIC model (Wang and Wang, 2016) was used to produce a simulation. The simulation provided the coefficient of deriving $O/N_2$ from a measured pair of 135.6 and LBH. The column O and $N_2$ ratio during the magnetic storm of Aug. 26, 2018 was presented in Fig. 11. On 24 August 2018 and most of 25 August 2018, Kp index was not more than 3. It abruptly rose to 7 in 26 August 2018. From 29 to 31 August 2018, Kp index was not more than 3. The column $O/N_2$ on 24 and 25 August was relatively quiet, and significant changes in column $O/N_2$ occurred on 26 and 27 August. The reduction of $O/N_2$ extended from the high-latitude region to mid- and low- latitude regions in the Northern and Southern Hemisphere. On 30 and 31 August, the column $O/N_2$ returned to quiet.**

**The column O and $N_2$ ratio derived from GUVI during the magnetic storm of Aug. 26, 2018 is presented in Fig. 12. The GUVI column $O/N_2$ data (Strickland et al., 2004) was obtained from GUVI website (http://guvitimed.jhuapl.edu/data_fetch _l3_on2_**

**idlsave).** **The column O/N$_2$ from GUVI on 24 and 25 August was relatively quiet, and significant changes in column O/N$_2$ occurred on 26 and 27 August. The reduction of O/N$_2$ also extended from the high-latitude region to mid- and low- latitude regions in the Northern and Southern Hemisphere. On 30 and 31 August, the column O/N$_2$ of GUVI also returned to quiet. The features of column O/N$_2$ of IPM and GUVI during the magnetic storm of Aug. 26, 2018 were similar. These results showed that the IPM data could provide a good monitoring of O/N$_2$ changes during the magnetic storm."]**

[Figure]

Figure 11: Column O/N$_2$ from IPM around the magnetic storm of Aug.26, 2018.

[Figure]

Figure 12: Column O/N$_2$ from GUVI around the magnetic storm of Aug.26, 2018.

2. Specific comments

- Are the IPM data open to public for an independent evaluation since that data in published papers are usually required to be accessible by public?

  **[Yes, the IPM data is available at http://satellite.nsmc.org.cn/PortalSite/Default.aspx.]**

- IPM calibration was done on ground. Was IPM calibrated in orbit?

**[No, the IPM aboard on the Feng Yun 3D meteorological satellite couldn't calibrated in orbit.]**

- Fig 2 (nightside channel). The non-zero responsivity around and below (likely) 130 nm suggests it is possible to pick up bright Lyman α emission around 121.6 nm. A discussion on this will be helpful. Fig 3. Dayside. Both bright 130.4 and 121.6 nm emissions could contribute the 135.6 channel.

**[A BaF$_2$ flat filter is used for short-wavelength cut, the transmittance shorter than 131nm is less than 2%, and it can cut 121.6nm radiation completely.]**

- Figure 4. The LBH band includes emission of N-1493, NO ε band, etc. Does the algorithm ignore the impact of none N2 LBH emissions?

**[At present our algorithm can't ignore the impact of none N$_2$ LBH emissions.]**

- Fig 5. It is necessary to show the data from the red leak channel. Furthermore, what is the responsivity of the red leak channel?

**[We added two panels in Fig. 5 in order to show the count of the red-leak channel. See Fig. 6 in the revised manuscript.]**

[Figure]

**Figure 6: The count of the 135.6nm nightside channel with red-leak (top), without red-leak (bottom), and the count of the red-leak nightside channel (middle) for new Moon (left) and full Moon (right) situation, respectively. March 17, 2018 is new Moon day, and March 31, 2018 is full Moon day.**

- Fig 6b. Change the color bar to show the equatorial arcs.

**[We have changed the color bar in Fig 6b. in order to show the equatorial arcs. See Fig. 7 in the revised manuscript.]**

[Figure]

**Figure 7: The global count (left) and brightness (right) of the 135.6nm nightside channel from 7 to 11 December 2017. The brightness is without red-leak and the effect of dark count. The red solid line indicates the magnetic dip equator.**

Is the SAA contamination removed in the same way as the red leak? If yes, why can it be done this way? If not, describe the method.

**[The dark count of the IPM is less than 10 per second generally. The SAA contamination makes the dark count increase significantly. Both the count of 135.6 nm nightside channel and the count of red-leak nightside channel include dark count contribution. The SAA contamination is removed by differencing measurements of 135.6 nm nightside channel and red-leak nightside channel.]**

- The plots show that data cover the entire Earth. Does the photometer scan in a cross track direction?

**[No, it doesn't. The plot is a merging map using five days data.]**

- Line 175. How can one estimate NmF2 based on the ratio between NmF2 and 135.6 nm intensity? It needs more details on the method.

**[We have reworded the sentence and added more details on the method. See Line 176-181 in the revised manuscript.]**

  **"The algorithm of deriving NmF$_2$ from the night time OI 135.6 nm emission is provided by Rajesh et al. (2011) and Jiang et al. (2014, 2018). The night time OI 135.6 nm emission is calculated based on nighttime OI 135.6 nm airglow radiative and emissive model. The electron density profile,the O$^+$ density profile,and the electron temperature profile are calculated using IRI2000 model,and the neutral components are calculated using MSISE90 model. The OI 135.6 nm emission is fitted to the square of NmF$_2$ linearly. The ratio of the square of NmF$_2$ to the OI 135.6 nm emission is obtained. Finally, NmF$_2$ is retrieved based on the observed OI 135.6 nm emission and the ratio."**

- Figure 8. The derived NmF2 should be plotted in the same format of Figure 6 to show the ionosphere morphology. It also needs a map of errors in the derived NmF2. What is the altitude of the Feng Yun 3-D? If the altitude is around NmF2 or above, the method wouldn't work.

**[The nighttime OI 135.6 nm airglow is proportional to the square of the maximum electronic density of ionospheric F$_2$ layer(NmF$_2$). The distribution of NmF$_2$ is similar to the distribution of the OI 135.6 nm airglow. We added a plot of the relative difference distribution (see Fig. 9 in the revised manuscript.) between IPM NmF$_2$ and IGGCAS ionosonde NmF$_2$. There is a standard deviation of 26.67% between IPM NmF$_2$ and IGGCAS ionosonde NmF$_2$.]**

**[The FY3D is an afternoon sun-synchronous satellite with an orbit altitude of 830 km. The algorithm of deriving NmF$_2$ from observed OI 135.6 nm emission is discussed by Rajesh et al (https://doi.org/10.1029/2010JA015686).]**

[Figure]

**Figure 9: The relative difference distribution between IPM NmF$_2$ and IGGCAS ionosonde NmF$_2$**

- Line 165. What is the local time of the observations in other studies?

**[The local time of GUVI on board TIMED satellite is near noon at the equator. The local time of the IPM orbit is about 2:00 am at the night-side equator.]**

- Line 180. How is the TEC estimated using the 135.6 nm radiance?

**[We added details on the method. See Line 176-181 in the revised manuscript.]**

**"The algorithm of deriving TEC from the night time OI 135.6 nm emission is provided by Rajesh et al. (2011) and Jiang et al. (2014). The process of deriving TEC based on the ratio between TEC and the night time OI 135.6 nm emission intensity is similar to that of deriving $NmF_2$".**

- Line 215. The net 135.6 nm and LBH radiances are estimated already. What is the reason to use a Butterworth filter to estimate the red leak due to cloud?

**[The IPM monitors 135.6 nm and $N_2$ LBH emissions in the day-side thermosphere by employing a filter wheel. The 135.6 nm dayside channel and the $N_2$ LBH dayside channel are asynchronous during observation. Fast varies from cloud tops could enter these channels, and make the count of these channels increase sharply. It leads to some of the $O/N_2$ features, such as sporadic enhanced $O/N_2$. We used a Butterworth filter in order to eliminate fast varies from cloud tops.]**

- This reviewer found a reference Wang and Wang 2016. It has the same title " Airglow simulation based on the Atmospheric Ultraviolet Radiance Integrated Code of 2012" and author names. Is this the same to the reference (Wang and Wang, 2015)?

  If they are the same paper, this reviewer couldn't find a AURIC based lookup table for the IPM O/N2 calculation.

**[Yes. The reference (Wang and Wang, 2016) provided an airglow simulation method based on the Atmospheric Ultraviolet Radiance Integrated Code. We used the method to obtain a lookup table that provides the coefficient of deriving $O/N_2$ from a measured pair of 135.6 and LBH values.]**

- Line 220. A plot of the O/N2 look up table should be added in the manuscript.

**[The manuscript focuses on the instrument validity of the IPM. The $O/N_2$ is provided only as a product sample. The process of deriving the O/N2 from the brightness of the O and $N_2$ emissions was discussed (Strickland et al., 1995; Zhang et al., 2004), and isn't the focal point of the manuscript. We did not describe it in detail in the manuscript. The following plot is our response to the referee.]**

[Figure]

- Since both 1356 and LBH channel include LBH, N-1493, NO ε bands. Are the contributions removed?

  **[At present our algorithm can't ignore them.]**

- Figure 9. How is the O/N2 data product validated? O/N2 depletion was seen on Aug 26, 2018 (storm-time). What are the sporadic enhanced O/N2 (vertical bars in red) over one or two orbits? Are they artifacts? If not, what cause the enhancements? What is the reference N2 column density for the O/N2 ratio?

  **[The sporadic enhanced $O/N_2$ (vertical bars in red) was originated from cloud tops reflection. The 135.6 nm dayside channel and the $N_2$ LBH dayside channel of the IPM are asynchronous during observation. Fast varies from cloud tops could enter these channels, and make the count of these channels increase sharply. It leads to some of the $O/N_2$ features, such as sporadic enhanced $O/N_2$. We optimized our filter algorithm and updated the result of $O/N_2$ product. And we also added the result of the $O/N_2$ from IPM compared with the $O/N_2$ from GUVI. Please refer to the first item of our response.]**

  **[The reference $N_2$ column density is $10^{17}$ $cm^{-2}$ for the $O/N_2$ ratio.]**

---

## Author Response (AR2)

**Response to the Associate Editor:**

Comments to the author:

Thank you for the updated version of the manuscript and for your responses to the reviewers' comments. The manuscript can now be accepted after some technical corrections.

Non-public comments to the Author:

Thank you for the updated version of the manuscript and for your responses to the reviewers' comments. Please implement the following corrections before the manuscript can be accepted:

**Answer: Thanks for the constructive comments. We have corrected them in our manuscript.**

- Line 35-36: Replace "in measuring" by "to measure".

  **[It is corrected according to the Associate Editor's comment. See Line 35-36 in the revised manuscript.]**

- Line 96: Replace "in ground" by "at ground".

  **[It is corrected according to the Associate Editor's comment. See Line 96 in the revised manuscript.]**

- Caption of figure 2: Replace "in ground" by "at ground".

  **[It is corrected according to the Associate Editor's comment. See Caption of figure 2 in the revised manuscript.]**

- Line 153: Replace "was" by "were".

  **[It is corrected according to the Associate Editor's comment. See Line 153 in the revised manuscript.]**

- Line 177: Replace "on nighttime" by "on a nighttime".

  **[It is corrected according to the Associate Editor's comment. See Line 177 in the revised manuscript.]**

- Line 178-179: Replace "using IRI2000 model" by "using the IRI2000 model".

  **[It is corrected according to the Associate Editor's comment. See Line**

178-179 in the revised manuscript.]

- Line 179: Replace "using MSISE90 model" by "using the MSISE90 model".

**[It is corrected according to the Associate Editor's comment. See Line 179 in the revised manuscript.]**

- Line 200: Replace "intergraded" with "integrated".

**[It is corrected according to the Associate Editor's comment. See Line 200 in the revised manuscript.]**

- Line 200: Replace "20200Km" with "20200 km". (Is the number correct?)

**[It is corrected according to the Associate Editor's comment. See Line 200 in the revised manuscript.]**

**[Yes, the number is correct.]**

- Line 201: Replace "intergraded" with "integrated".

**[It is corrected according to the Associate Editor's comment. See Line 201 in the revised manuscript.]**

- Line 201: Replace "830Km" with "830 km".

**[It is corrected according to the Associate Editor's comment. See Line 201 in the revised manuscript.]**

- Line 229: Remove "of".

**[It is corrected according to the Associate Editor's comment. See Line 229 in the revised manuscript.]**

- Line 230: Replace "The retrieval algorithm could refer in relevant paper (Strickland et al., 1995; Zhang et al., 2004)." by "The retrieval algorithm was decribed by Strickland et al. (1995) and Zhang et al. (2004)."

**[It is corrected according to the Associate Editor's comment. See Line 230 in the revised manuscript.]**

- Line 231: Replace "The brightness of 135.6 nm emission and N2 LBH emission on dayside" by "The brightness of the 135.6 nm emission and the N2 LBH emission on the dayside".

**[It is corrected according to the Associate Editor's comment. See Line 231 in the revised manuscript.]**

- Line 232: Replace "deducting" by "deduct".

  **[It is corrected according to the Associate Editor's comment. See Line 232 in the revised manuscript.]**

- Line 233: Replace "in data processing" by "in the data processing".

  **[It is corrected according to the Associate Editor's comment. See Line 233 in the revised manuscript.]**

- Line 234: Replace "of deriving" by "for deriving".

  **[It is corrected according to the Associate Editor's comment. See Line 234 in the revised manuscript.]**

- Line 234: Replace "135.6" by "135.6 nm".

  **[It is corrected according to the Associate Editor's comment. See Line 234 in the revised manuscript.]**

- Line 236: Replace "rose 7" by "rose to 7".

  **[It is corrected according to the Associate Editor's comment. See Line 236 in the revised manuscript.]**

- Line 241: Replace "from GUVI website" by "from the GUVI website".

  **[It is corrected according to the Associate Editor's comment. See Line 241 in the revised manuscript.]**